# The Effect of Von Willebrand Disease on Pregnancy, Delivery, and Postpartum Period: A Retrospective Observational Study

**DOI:** 10.3390/medicina58060774

**Published:** 2022-06-07

**Authors:** Mateja Sladič, Ivan Verdenik, Špela Smrkolj

**Affiliations:** 1Division of Gynaecology and Obstetrics, University Medical Centre, 1000 Ljubljana, Slovenia; sladic.mateja@gmail.com (M.S.); ivan.verdenik@guest.arnes.si (I.V.); 2Faculty of Medicine, University of Ljubljana, 1000 Ljubljana, Slovenia

**Keywords:** von Willebrand disease, pregnancy, delivery, postpartum hemorrhage

## Abstract

*Background and Objectives* Several reports indicate that women with von Willebrand disease (VWD) are at an increased risk of bleeding and other complications during pregnancy and childbirth. The aim of this study was to investigate the effect of VWD on the course of pregnancy, childbirth, and the postpartum period. *Materials and Methods* This was a retrospective study that compared many variables between women with VWD (*n* = 26) and women without VWD (*n* = 297,111) who gave birth between 2002 and 2016 in Slovenia. Data were obtained from the Slovenian National Perinatal Information System. *Results* Women with VWD were not more likely to have a miscarriage, vaginal bleeding during pregnancy, anemia, intrauterine growth restriction, or imminent premature labor. However, women with VWD were more likely to experience childbirth trauma-related bleeding (OR, 10.7; 95% CI: 1.4, 78.9), primary postpartum hemorrhage (OR, 3.7; 95% CI: 0.9, 15.8), and require blood transfusion after childbirth (OR, 16.3; 95% CI: 2.2, 120.3). No cases of stillbirth or early neonatal death were observed in women with VWD. *Conclusion* Although women with VWD did not demonstrate an increased risk of vaginal bleeding during pregnancy or poor fetal outcomes, they had a higher risk of primary postpartum hemorrhage and requiring blood transfusion.

## 1. Introduction

Von Willebrand disease (VWD), first described by Dr. Erik von Willebrand, is the most common autosomal inherited bleeding disorder. It is caused by deficiency or dysfunction of von Willebrand factor (VWF), a plasma protein that mediates the initial adhesion of platelets at sites of vascular injury, and binds and stabilizes blood clotting factor VIII (FVIII) in the circulation. As such, defects in VWF can lead to bleeding because of impaired platelet adhesion or reduced FVIII concentrations. The prevalence of VWD varies between different studies and depends on the definition that is used, as not all VWD patients develop bleeding symptoms. It is estimated that VWD affects 0.5–1.3% of the population [1,2,3].

VWD is classified into three major categories. Type 1 is the most common (partial quantitative deficiency) and accounts for more than 80% of cases. Type 2 (qualitative deficiency) is further divided into four subtypes (2A, 2B, 2M, and 2N), according to different impairments in the form and/or function of VWF, and accounts for 15–20% of cases. Type 3 corresponds to an almost total absence of VWF and is the most rare, with less than 1% of cases [3,4].

The most common presentations of VWD are minor mucocutaneous bleeding, such as epistaxis, gingival bleeding, or easy bruising. Some patients may also report prolonged postoperative bleeding after wisdom tooth extraction or tonsillectomy. Severe presentations, such as gastrointestinal bleeding or hemarthrosis, are rare and are usually associated with type 2 or 3 VWD [5,6]. Although VWD occurs equally in males and females, women are at a higher risk of being affected, mostly because of bleeding challenges associated with menstruation, pregnancy, and childbirth [1,7,8].

During pregnancy, many changes in hemostasis occur that result in a hypercoagulable state. The levels of several hemostatic factors increase, including factors VII and X, fibrinogen, and plasminogen activator inhibitor type 1. Conversely, the levels of anticoagulant factors, such as protein S, decrease. Furthermore, FVIII and VWF levels change during pregnancy in both women with and without VWD [9]. The levels of both factors start increasing in the second trimester and peak during the third trimester; these increases depend on the type and subtype of VWD. On the other hand, women with type 1 and 2 VWD usually achieve normal VWF and FVIII levels at the end of pregnancy; these levels are unchanged in women with type 3 VWD during pregnancy [5]. FVIII and VWF levels decrease rapidly after delivery in women with VWD, approaching baseline after 1 week and reaching baseline after 3 weeks. Consequently, women with VWD may be at risk of postpartum hemorrhage [10].

In Slovenia, the Registry of Inherited Bleeding Disorders was established in 1967 and has existed as a computerized database since 1998 [11]. In 2018, 187 women with inherited bleeding disorders were registered, of which 127 had confirmed VWD. As VWD is a relatively rare bleeding abnormality and the population of Slovenia is small, the data are still limited in Slovenia. To the best of our knowledge, this is the first study in Slovenia to explore the effect of VWD on the course of pregnancy, delivery, and the postpartum period.

## 2. Materials and Methods

The design of this study is retrospective and observational. Maternal characteristics of pregnancies and various obstetrics complications as well as neonatal outcomes were collected using the Slovenian National Perinatal Information System (NPIS) data collecting protocol. NPIS registers all deliveries in Slovenia at ≥22 weeks gestation or when the fetus weighs ≥ 500 g. Registration is mandatory by law, and more than 140 variables are entered immediately postpartum into a computerized database by the attending midwife and physician after delivery. Patient demographics, family, medical, gynecological and obstetric history, data on current pregnancy, labor and delivery, postpartum period and neonatal outcomes are collected. The complete list of variables with definitions and methodological guidelines has been published online by the Slovenian Institute of Public Health [12]. For the purpose of ensuring the quality of data entered, automatic error reports are built into the computerized system. Moreover, the data are audited periodically and compared to those from international databases, such as the Vermont Oxford network, in which Slovenia participates.

The study group represented women with VWD (*n* = 26) who gave birth in Ljubljana maternity hospital between 2002 and 2016. Among women with VWD, there were 23 women with confirmed type 1 VWD, 2 women with type 2 VWD and 1 woman with type 3 VWD. The control group (*n* = 297,111) included women who gave birth in any maternity hospital in Slovenia during the same period as the study group. The following variables were compared between the groups: maternal age, gestational age at birth (weeks), incidence of obstetric complications, e.g., previous miscarriage, vaginal bleeding during pregnancy, anemia, thrombocytopenia, cerclage, fetal growth restriction (defined as estimated fetal weight (EFW) < 5th percentile, or abdominal circumference < 5th percentile combined with oligohydramnios (amniotic fluid index-AFI < 5 cm), and/or an umbilical artery PI > 95th percentile), imminent premature labor, and elective and total caesarean sections. Furthermore, we evaluated the incidence of various complications at the time of birth and during the postpartum period: primary postpartum hemorrhage (defined as a loss of > 500 mL of blood within 24 h after birth), childbirth trauma-related bleeding (defined as bleeding due to cervical, vaginal and perineal lacerations or uterine rupture that occur during vaginal delivery and exceed 500 mL), secondary postpartum hemorrhage (defined as bleeding that exceeds normal lochial loss 24 h to 6 weeks after delivery), postpartum anemia, and received blood transfusions (in our institution, we register transfusions of all blood products as blood transfusions, which include packed red blood cells, platelets, fresh frozen plasma and cryoprecipitate).

Data were also obtained regarding neonatal outcomes, including the 5 min “Appearance, Pulse, Grimace, Activity, and Respiration” (APGAR) score of infants and the incidence of stillbirths and neonatal deaths.

We used the Statistical Package for the Social Sciences (SPSS software, version 27, Chicago, IL, USA) for analysis. Student’s t-test was used for normal variables, the Mann–Whitney *U* test for nonparametric, and χ2-test for categorical variables. *p* < 0.05 was considered statistically significant. The odds ratio (OR) and 95% confidence interval (CI) were calculated. This retrospective study of anonymous entries was exempt from approval by the local ethical committee.

## 3. Results

Between 2002 and 2016, 297.137 women gave birth in Slovenian maternity hospitals, among which 26 (0.0087%) women had a diagnosis of VWD. Our analysis reveals a similar maternal age, parity, and gestational age at birth between the study and control groups. Furthermore, women with VWD were not more likely to have cervical cerclage, anemia, or thrombocytopenia than women without VWD (Table 1).

Additionally, there were no cases of intrauterine fetal growth restriction in our study group. Women with VWD had a similar incidence of previous miscarriages as women without VWD. Women with VWD were not more likely to experience vaginal bleeding in the first and second trimester of pregnancy, but were five times more likely to experience vaginal bleeding in the third trimester of pregnancy; however, the data did not reach statistical significance (OR, 5.6; 95% CI: 0.8, 41.4). Although vaginal bleeding during pregnancy is a well-known risk factor for preterm labor, the incidence for imminent premature labor was comparable between both groups. Additionally, the incidence of cesarean sections was similar in both groups.

We found that women with VWD were more likely (OR, 3.7; 95% CI: 0.9, 15.8) to experience primary postpartum hemorrhage (7.7%) than women without VWD (2.2%). Moreover, women with VWD showed a significantly higher incidence of childbirth trauma-related bleeding (OR, 10.7; 95% CI: 1.4, 78.9) and were more likely to require a blood transfusion within the first 24 h after childbirth (OR, 16.3; 95% CI: 2.2, 120.3). Both groups exhibited a similar incidence of postpartum anemia. Women with VWD had no cases of third or fourth degree perineal tears, secondary postpartum hemorrhage, or need for blood transfusion more than 24 h after childbirth.

We found that infants born to women with VWD had an equal median APGAR score at 5 min (9; IQR 9–9) compared to controls (9; IQR 9–10) (Table 2). Moreover, women with VWD had no cases of stillbirths or neonatal deaths.

## 4. Discussion

VWD is the most common inherited hemostatic disorder. The last 20 years have witnessed progress in understanding, diagnosing, and treating bleeding complications in pregnant women with VWD [9]. Moreover, for almost three decades in Slovenia, all pregnant women with VWD were managed with a tight collaboration between obstetricians and hematologists. The latter regularly monitor coagulation factor levels, and promptly and carefully evaluate pregnant women with VWD [13]. Altogether, these approaches are needed to form customized treatment plans at childbirth. In our institution, obstetricians collaborate with hematologists regarding the administration of prophylactic anti-hemorrhagic treatment in women with VWD during pregnancy, labor and the postpartum period. In doing so, we rely on the World Federation of Hemophilia’s guidelines [14]. Women with VWD have levels of blood clotting factor VIII (FVIII), von Willebrand factor (VWF:Ag), and von Willebrand factor ristocetin cofactor activity (VWF:RCo) measured at the time of delivery or at least two weeks before expected delivery, and more frequently if complications occur. Additionally, we decide on the possible treatment options depending on the type of VWD. Women with mild type 1 VWD rarely require treatment. The risk of postpartum hemorrhage is low if the levels of FVIII are above 0.40 IU/L before delivery. Consequentially, in these cases, no prophylactic treatment is needed. Otherwise, if the levels of FVIII are below this value, prophylactic treatment with desmopressin (DDAVP) is initiated. DDAVP is a first-line prophylactic treatment for women with VWD type 1 in our institution. On the contrary, it is inefficient in women with type 3 VWD and in most women with type 2 VWD. Moreover, it is contraindicated in women with type 2B VWD, as it can lead to thrombocytopenia. Plasma-derived VWF and FVIII concentrates such as Wilate^®^ and Hemate P^®^ are the treatment options for women with type 3 VWD, type 2B VWD and types 1 and 2 VWD who do not respond to DDAVP. The balance of VWF:RCo and FVIII is 1:1 in Wilate^®^ concentrate and 2.4:1 in Hemate P^®^ concentrate. The starting dose is somewhere between 20 and 50 IU/kg. Before delivery, and until the 5th day after the delivery, FVIII values are maintained above 0.40 iU/L. Other treatment options represent antifibrinolytics (e.g., tranexamic acid) and transfusion of platelets [13,14].

In this retrospective study, we investigated the correlation between VWD and the incidence of various complications during pregnancy, childbirth, and the postpartum period.

Our study revealed that women with VWD were slightly, but not significantly, older than women without VWD (30.1 ± 5.6 vs. 29.5 ± 4.8 years). One possible explanation for this could be that women with VWD are concerned about their condition and possible bleeding complications and, thus, postpone pregnancy. Similar data were reported by James et al. in the USA, where women with VWD were older than women without VWD at childbirth [15].

Generally, it is known that women with VWD are more prone to developing anemia and thrombocytopenia during pregnancy, which can be explained by their bleeding tendency [16]. The previously mentioned study concluded that women with VWD were two-fold more likely to experience anemia and thrombocytopenia during pregnancy than women without VWD [15]. Conversely, our analysis presented no case of thrombocytopenia and only one case of anemia during pregnancy in women with VWD. This could be due to the small population in Slovenia (two million people) and the consequently small number of pregnant women with VWD (*n* = 26). Nonetheless, the multidisciplinary approach to managing our patients, which includes a hematologist and obstetrician, could play an important role in the lower incidence of anemia and thrombocytopenia in our group of women with VWD. Namely, for each woman with VWD, the hematologist prepares a customized plan for prophylaxis and/or treatment if needed during pregnancy, childbirth, or the postpartum period. Such plans can be adjusted depending on different laboratory measurements of hemostatic parameters and possible bleeding complications. Lastly, all pregnant women with VWD in Slovenia are managed in one national hemophilia center.

Our data demonstrated that women with VWD were not more likely to develop intrauterine fetal growth restriction. This is not surprising, as it is very unlikely that VWD could influence fetal growth. In addition, two other studies observed similar data. Skeith et al. found that the incidence of placenta-mediated complications, including fetal growth restriction, did not differ between women with VWD and women without VWD [17]. James et al. also found that the incidence of fetal growth restriction was similar between women with VWD and women without VWD [15].

Furthermore, women with VWD were not more likely to experience vaginal bleeding in the first and second trimester of pregnancy. Additionally, women with VWD had a similar incidence of miscarriages. These results are expected, as bleeding in pregnancy is a well-known risk factor for miscarriage. However, Kadir et al. demonstrated high rates of bleeding in the first trimester for women with VWD (33%). Nevertheless, the miscarriage rates were not increased (21%) [18]. The background risk for miscarriage varies depending on the data source and is estimated to occur in up to 25% of clinically recognized pregnancies in developed countries [19,20].

In the literature, antepartum bleeding (defined as bleeding in the third trimester) has been recognized as a predisposing factor for preterm labor [21,22]. In our study, women with VWD were five-fold more likely to experience antepartum bleeding; however, these results were not statistically significant. Moreover, no association between imminent premature labor and VWD was observed. This observation is in agreement with that of James et al., who concluded that women with VWD were not more likely to experience preterm labor than women without VWD. Nevertheless, their study demonstrated that women with VWD were ten-fold more likely to experience antepartum bleeding [15].

The type of anesthesia used during delivery in patients with VWD is controversial. Patients with VWD are at an increased risk of developing a hematoma as a result of the use of epidural analgesia [23]. The use of factor replacement products at low levels of VWF or FVIII may reduce the risk of bleeding with epidural analgesia [4,5]. In our institution, we collaborate closely with anesthesiologists to ensure that anesthesia and labor analgesia are administered safely and effectively. Because of the higher risk of bleeding complications (e.g., hematoma), limited reports regarding the safety of neuraxial anesthesia (lumbar epidural and combined spinal-epidural analgesia) and no definitive guidelines for labor analgesia in women with VWD, in our institution, we mostly do not perform neuraxial analgesia in those patients. The decision to use neuraxial analgesia has to be thoughtfully considered, and could be a possible option for labor analgesia for women with type 1 VWD with recommended levels of VWF and FVIII > 0.50 IU/L before the procedure. Moreover, in our tertiary center, other pharmacological options for labor analgesia are offered as well, such as inhalation analgesia, where a mixture of 50% nitrous oxide and 50% oxide is used. Another option is patient-controlled intravenous opioid analgesia with remifentanil used as an active substance. Additionally, we advise patients to avoid drugs that have a potential impact on hemostasis, including non-steroidal anti-inflammatory agents. These medications, which are regularly prescribed for pain release in the postpartum period, may adversely affect platelet function and increase the risk of bleeding.

During pregnancy, several hemostatic changes occur, including increased levels of coagulation factor VIII and VWF, which play a protective role and contribute to improved hemostasis. However, women with VWD have lower levels of these factors and a higher incidence of postpartum hemorrhage [2,24]. There are conflicting results in the literature regarding the risk and severity of bleeding during childbirth in women with VWD. Interestingly, in a historical study by Silwer, postpartum hemorrhage was almost as common among women with VWD (23.5%) as among women without VWD (19.5%) [8]. The latter could be explained by differences in the parity, as the majority of the women without VWD were parous. Of parous women with VWD, in that same study, the frequency of postpartum hemorrhage was 34.5% [8]. Another case–control study by Kirtava et al. from “Centers for Disease Control and Prevention” reported a much higher incidence of postpartum hemorrhage in women with VWD (59%) compared to those without VWD (21%) [25]. Our data showed that women with VWD were more likely to experience postpartum hemorrhage; however, the odds (3.7; 95% CI: 0.9, 15.8) were not as high as expected. Furthermore, all the women included in our study group were previously diagnosed with VWD, which enabled obstetricians to anticipate possible bleeding complications and provide timely prophylactic anti-hemorrhagic treatment in agreement with hematologists.

Although the numbers were small, based on our results, women with VWD were ten-fold more likely to experience childbirth trauma-related bleeding, with an incidence of 3.8%. This is in accordance with a study by Kadir et al., who reported that 3 out of 49 women with VWD and vaginal deliveries experienced extensive perineal bruising and hematoma [18].

Despite the increased awareness of bleeding tendencies in women with VWD and the remarkable improvement in clinical care and guidelines, postpartum hemorrhage remains an important concern in these women [1,26]. Moreover, despite several prophylactic anti-hemorrhagic treatment options being available nowadays, such as 1-deamino-8-D-arginine vasopressin and FVIII/VWF concentrate, no significant decline in postpartum hemorrhage has occurred over the last decades. In a survey of 423 women with VWD, published by De Wee et al., 11% of women received blood transfusion after delivery (vaginal and caesarean included) [1]. This incidence is much higher than that of the general population, in which only 1% of women need blood transfusion after vaginal delivery and 1–7% after caesarean section [27]. Similarly, our study showed that women with VWD were more likely to receive blood transfusions less than 24 h after delivery (vaginal and caesarean included) compared with women without VWD. However, the incidence was lower than expected (3.8% vs. 0.2% for women with VWD and women without VWD, respectively).

In normal circumstances, postpartum hemorrhage is controlled by progressively increased VWF and FVIII levels, especially in the third trimester of pregnancy. VWF and FVIII levels peak at the time of delivery and decline afterwards, returning to baseline within 1 month postpartum [10,28]. Interestingly, James et al. demonstrated that the peak in VWF levels occurred 12 h postpartum in women without VWD and 4 h in women with VWD. Furthermore, VWF levels declined rapidly after childbirth, approaching baseline after 1 week in women with VWD [10]. These findings may indicate a risk for secondary postpartum hemorrhage. In the literature, the incidence of secondary postpartum hemorrhage in women with VWD varies from 20% [18] to 25% [29] and 28% [30]. These rates are much higher than the expected background rate for secondary postpartum hemorrhage, which is less than 1% [24]. Conversely, in our study, none of the women with VWD suffered secondary postpartum hemorrhage, compared to 0.2% of women without VWD. Additionally, there was no case of blood transfusion received more than 24 h after childbirth among women with VWD. Lastly, women with VWD were not more likely to develop postpartum anemia compared to women without VWD. It should be noted that the numbers in our study were small. However, we assume that the multidisciplinary approach to treating women with VWD (involving a hematologist and obstetrician, regular measurements of VWF and FVIII levels postpartum, and appropriate treatment) contributes to the lower incidence of bleeding complications in the late postpartum period.

In our study, we focused on the effect of VWD on several neonatal outcomes. Delivery presents a critical event not only for women with VWD, but also for their affected infants. Because the infant is at potential risk of serious complications, e.g., scalp and intracranial hemorrhage during childbirth and instrumental deliveries, it is recommended that delivery should be achieved in the least traumatic manner [18]. In our study, there were no cases of stillbirth or early neonatal death among infants born to women with VWD. The median APGAR score at 5 min was equal for infants born to women with VWD (9; IQR 9-9) and women without VWD (9; IQR 9-10). This finding is in agreement with an observational study by Wilson et al. from Australia, who only included women with VWD (23 deliveries overall) and reported a median APGAR score at 5 min of nine [31].

In Slovenia, all children born to women with VWD are screened for these inherited disorders later on during their first year of life, because clotting factors reach adult levels at the age of 6 months. As previously mentioned, VWD is autosomal inherited, and, thus, newborns can be affected by this disorder as well [18]. Thus, it is necessary to determine the diagnosis of VWD early in life, as this enables complete management by a hematologist, and prevention and treatment of possible future bleeding events.

The main limitation of our study is the small study group. Nevertheless, it must be taken into consideration that Slovenia is a country with a population of only two million people, and that we included all women with VWD that gave birth within a period of 15 years. The potential strength of our study is that it emphasizes collaborative multidisciplinary approaches, which include both obstetricians and hematologists and individualized management of women with VWD in tertiary specialized referral centers.

## 5. Conclusions

In conclusion, there is a higher incidence of particular complications during childbirth and the postpartum period in women with VWD, such as primary postpartum hemorrhage, childbirth trauma-related bleeding and receiving blood transfusion within the first 24 h after childbirth. Increased awareness and a better understanding of this bleeding disorder and strong interdisciplinary collaborations are of utmost importance in achieving optimal outcomes and minimizing maternal and neonatal complications.

## Figures and Tables

**Table 1 medicina-58-00774-t001:** Maternal characteristics of pregnancies and the odds of obstetric complications in women with von Willebrand disease (VWD) compared with women without VWD.

Total	With VWD	Without VWD	Odds Ratios with 95% CI	*p*-Value
*n* = 287,137	*n* = 26	*n* = 297,111		
Mean age (years)	30.1 ± 5.6	29.5 ± 4.8	-	0.53
Nulliparous	13 (50.0)	147,052 (49.5)	1.0 (0.5, 2.2)	0.96
Mean gestational age (weeks)	39 (38–40)	39 (38–40)	-	0.093
Anemia	1 (3.8)	9315 (3.1)	1.2 (0.2, 9.1)	0.84
Thrombocytopenia	0 (0)	536 (0.2)	-	0.83
Cerclage	0 (0)	1323 (0.4)	-	0.73
Intrauterine growth restriction	0 (0)	8243 (2.8)	-	0.39
Previous miscarriage	6 (23.1)	49,145 (16.5)	1.5 (0.6, 3.8)	0.37
Vaginal bleeding during pregnancy:				
I. trimester	3 (11.5)	16,377 (5.5)	2.2 (0.6, 7.4)	0.18
II. trimester	1 (3.8)	3810 (1.3)	3.1 (0.4, 22.7)	0.25
III. trimester	1 (3.8)	2105 (0.7)	5.6 (0.8, 41.4)	0.06
Imminent premature labor	2 (7.7)	12,043 (4.1)	2.0 (0.5, 8.3)	0.35
Elective cesareans	2 (7.7)	21,812 (7.3)	1.1 (0.3, 4.5)	0.95
Total cesareans	3 (11.5)	51,394 (17.3)	0.6 (0.2, 2.1)	0.44
Primary postpartum hemorrhage	2 (7.7)	6503 (2.2)	3.7 (0.9, 15.8)	**0.05**
3rd and 4th degree perineal tears	0 (0)	950 (0.3)	-	0.77
Uterine atony	0 (0)	5163 (1.7)	-	0.50
Childbirth trauma-related bleeding	1 (3.8)	1109 (0.4)	10.7 (1.4, 78.9)	**0.004**
Blood transfusion ≤ 24 h after childbirth	1 (3.8)	728 (0.2)	16.3 (2.2, 120.3)	**0.000**
Postpartum anemia	4 (15.4)	36,839 (12.4)	1.3 (0.4, 3.7)	0.64
Secondary postpartum hemorrhage	0 (0)	259 (0.1)	-	0.88
Blood transfusion > 24 h after childbirth	0 (0)	1871 (0.6)	-	6.85

Data are shown as mean ± SD or n (%) or median/interquartile range (q3–q1). Significant values are shown in bold.

**Table 2 medicina-58-00774-t002:** Neonatal outcomes of infants born to mothers with von Willebrand disease (VWD) in comparison to mothers without VWD.

	With VWD	Without VWD	*p*-Value
APGAR score at 5 min	9 (9–9)	9 (9–10)	**0.009**
Stillbirth	0 (0)	1641 (0.5)	0.71
Early neonatal death	0 (0)	516 (0.2)	0.83

Data are shown as mean ± SD or n (%) or median/interquartile range (q3-q1 Significant values are shown in bold.

## Data Availability

The data that support the findings of this study are available from the corresponding author, Š.S., upon reasonable request.

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
