# Peer review of "The Effect of Von Willebrand Disease on Pregnancy, Delivery, and Postpartum Period: A Retrospective Observational Study"

_medicina, 2022, doi:10.3390/medicina58060774_

Round 1
Reviewer 1 Report
This is an interesting study in Slovenia to explore the effect of Von Willebrand Disease during pregnancy on the course of pregnancy, delivery, and the postpartum period. Defects in VWF can lead to bleeding because of impaired platelet adhesion or reduced FVIII concentrations and this is a challenge in obstetrics care.
Objectives: The text is clear and objectives are clear and well designed.
Methods:
The inclusion criteria were presented stating that the data come from a national registry. Could the authors clarify and provide details on how cases are recorded in the system? Inform if this occurs after delivery and who is responsible for the registration.
Could the authors clarify whether institutions follow a specific protocol that provides criteria for prophylactic infusion of factor VIII, before or after delivery, for patients with Von Willebrand disease?
Anesthetic procedures are challenging in these cases. Could the authors clarify the management protocol for anesthesia or labor analgesia in von Willebrand's disease?
Could you clarify how bleeding related to major trauma was characterized?
The study report on blood transfusion within 24 hours of birth, but it is not clear whether this includes the transfusion of blood products (plasma, cryoprecipitate, platelets, etc.)?
Could you clarify the criterion used to characterize fetal growth restriction?
In the description of the statistical analysis, please describe the tests used for normal and non-parametric variables.
Results:
The results of Apgar scores (I understand that it is not an acronym, since it was first proposed by Dr Virginia Apgar, who developed the score) are usually non-parametric, and therefore it would be better to present medians and IQR.
Discussion:
They could clarify whether there was an analysis of birth weight and the occurrence of small-for-gestational-age newborns.
In general, the discussion is well presented and the conclusions are relevant.
Author Response
We thank the reviewer for her/his valuable insights and constructive comments that have contributed to a better quality of our manuscript. Below are our responses in italics:
Responses to reviewer 1
This is an interesting study in Slovenia to explore the effect of Von Willebrand Disease during pregnancy on the course of pregnancy, delivery, and the postpartum period. Defects in VWF can lead to bleeding because of impaired platelet adhesion or reduced FVIII concentrations and this is a challenge in obstetrics care.
Objectives: The text is clear and objectives are clear and well designed.
Methods:
The inclusion criteria were presented stating that the data come from a national registry.
- Could the authors clarify and provide details on how cases are recorded in the system? Inform if this occurs after delivery and who is responsible for the registration.
Maternal characteristics of pregnancies and various obstetrics complications as well as neonatal outcomes were collected using the National Perinatal Information System (NPIS) data collecting protocol. NPIS registers all deliveries in Slovenia at ≥ 22 weeks’ gestation or when the fetus weights ≥ 500 g. Registration is mandatory by law and more than 140 variables are entered into a computerized database by the attending midwife and physician after delivery. Patient demographics, family, medical, gynecological and obstetric history, data on current pregnancy, labor and delivery, postpartum period and neonatal outcomes are collected. The complete list of variables with definitions and methodological guidelines has been published on-line by the Slovenian Institute of Public Health (https://www.nijz.si/sites/www.nijz.si/files/uploaded/podatki/podatkovne_zbirke_raziskave/pis/peris-metodoloska-navodila-2022-v2-4.pdf). For the purpose of ensuring the quality of data entered, automatic error reports are built into the computerized system. Moreover, the data is audited periodically and compared to those from international databases, such as Vermont Oxford network, in which Slovenia participates.
We added these information and additional reference to section 2. (Materials and methods).
- Could the authors clarify whether institutions follow a specific protocol that provides criteria for prophylactic infusion of factor VIII, before or after delivery, for patients with Von Willebrand disease?
Thank you for your question. As it was already mentioned in the article, in our institution obstetricians collaborate with hematologists regarding administration of prophylactic anti-hemorrhagic treatment in women with VWD during pregnancy, labor and postpartum period. In doing so, we rely on the World Federation of Hemophilia’ guidelines (https://wfh.org/). Women with VWD have levels of blood clotting factor VIII (FVIII), von Willebrand factor (VWF:Ag), von Willebrand factor ristocetin cofactor activity (VWF:RCo) measured at the time of delivery or at least two weeks before expected delivery and more frequently if complications occur. Additionaly, we decide on the possible treatment options depending on the type of VWD. Women with mild type 1 of VWD rarely require treatment. The risk of postpartum hemorrhage is low if the levels of FVIII are above 0,40 IU/L before delivery. Consequentially in these cases no prophylactic treatment is needed. Otherwise, if the levels of FVIII are below this value, prophylactic treatment with desmopressin (DDAVP) is initiated. DDAVP is a first-line prophylactic treatment for women with VWD type 1 in our institution. On the contrary, it is inefficient in women with type 3 VWD and most women with type 2 VWD. Moreover it is contraindicated in women with type 2B VWD, as it can lead to thrombocytopenia.
Plasma derived VWF and FVIII concentrate such as Wilate® and Hemate P® are the treatment options for women with type 3 VWD, type 2B VWD and types 1 and 2 VWD who do not respond to DDAVP. The balance of VWF:RCo and FVIII, is 1:1 in Wilate® concentrate and 2,4:1 in Hemate P® concentrate. The starting dose is somewhere between 20 to 50 IU/kg. Before delivery and untill 5th day after the delivery FVIII values are mainted above 0,40 iU/L. Other treatment option represent antifibrinolytics (e.g. tranexamic acid) and transfusion of platelets.
We added these information and additional reference to section 4. (Discussion).
- Anesthetic procedures are challenging in these cases. Could the authors clarify the management protocol for anesthesia or labor analgesia in von Willebrand's disease?
We agree with your opinion regarding difficulties that anesthetic procedures represent for women with VWD during labor and postpartum period. In our institution we collaborate closely with anesthesiologist to ensure safe and effective anesthesia and labor analgesia. Because of higher risk of bleeding complications (e.g. hematoma), limited reports regarding safety of neuraxial anesthesia (lumbar epidural and combined spinal-epidural analgesia) and no definitive guidelines for labor analgesia in women with VWD, in our institution we mostly do not perform neuraxial analgesia in those patients. The decision for use of neuraxial analgesia has to be thoughtfully considered and could be a possible option of labor analgesia for women with type 1 VWD with recommended levels of VWF and FVIII > 0,50 IU/L before procedure. Moreover, in our tertiary centre other pharmacological options for labor analgesia are offered as well, such as inhalation analgesia, where a mixture of 50 % nitrous oxide and 50 % oxyde is used. Another option is patient-controlled intravenous opioid analgesia with remifentanil used as an active substance.
Additionaly, we advise patients to avoid drugs that have a potential impact on hemostasis, including non-steroidal anti-inflammatory agents. This medication, which are regulary prescribed for pain release in postpartum period, may adversely affect platelet function and increase risk of bleeeding.
We added these information and additional references to section 4. (Discussion).
- Could you clarify how bleeding related to major trauma was characterized?
Thank you for your question. Childbirth trauma-related bleeding is characterized as bleeding due to cervical, vaginal and perineal lacerations or uterine rupture that occur during vaginal delivery and exceeds 500 ml.
We added this information to section 2. (Materials and methods).
- The study report on blood transfusion within 24 hours of birth, but it is not clear whether this includes the transfusion of blood products (plasma, cryoprecipitate, platelets, etc.)?
Thank you for your observation. In our institution we register under blood transfusions transfusions of all blood pruducts, which includes packed red blood cells, platelets, fresh frozen plasma and cryoprecipitate.
We added this information to section 2. (Materials and methods).
- Could you clarify the criterion used to characterize fetal growth restriction?
Thank you for pointing out missing data about criticria used to characterize fetal growth restriction (FGR) in our manuscript. FGR was defined as estimated fetal weight (EFW) < 5th percentile, or abdominal circumference < 5th percentile combined with oligohydramnios (amniotic fluid index-AFI < 5 cm), and/or an umbilical artery PI > 95th percentile.
We added this information to section 2. (Materials and methods).
- In the description of the statistical analysis, please describe the tests used for normal and non-parametric variables.
Thank you for reminding us of nonstated tests used for statistic analysis in our manuscript. Missing informations were added to section 2. (Materials and methods).
- Results:
The results of Apgar scores (I understand that it is not an acronym, since it was first proposed by Dr Virginia Apgar, who developed the score) are usually non-parametric, and therefore it would be better to present medians and IQR.
Thank you for your observation and thoughtful suggestions. We agree with your opinion about Apgar score, as it was originally developed by an anesthesiologist Dr. Virginia Apgar to evalute newborns shortly after birth. Apgar score is actually a backronym of Dr. Apgar's name, which means that it is formed of already existing names. With help of this score we evaluate five criteria of the newborn: activity, pulse, grimace, appearance and respiration. Additionaly, we appreciate your insight and suggestion regarding results of Apgar score. As it is a nonparametric variable we corrected statistic analysis and for this purpose we used Mann-Whitney U test, the results are presented as median and interquartile range (IQR).
- Discussion:
They could clarify whether there was an analysis of birth weight and the occurrence of small-for-gestational-age newborns. In general, the discussion is well presented and the conclusions are relevant.
Thank you for your approval and well opinion about our manuscript. In our study we have not performed analysis of birth weight of infants born to mothers with VWD and those without VWD, nor there was any analysis of small-for gestiational-age newborns. There was only made comparison between both groups of women regarding incidence of intrauterine growth restriction.
Reviewer 2 Report
I thought this was a very valuable paper summarizing the perinatal management of pregnancies complicated by vWD in Slovenia. The content is quite interesting, but I think that the statistical analysis should use medians rather than means, as there must be a large skew in the distribution of the resulting data.
1. Most of the Apgar scores, for example, would be 9 or 10. So, I would think that there would be no significant difference? Also, in Table 1, the presence of postpartum hemorrhage, blood transfusion, and bleeding due to perineal laceration are all correlated, which would explain the same thing, wouldn't it? We believe that the degree of anemia and platelet counts prior to delivery affect the postpartum values. Please reconsider this information as it affects the results of the paper.
2. Please re-consider what might be prognostic factors in a patient with vWD.
Author Response
We thank the reviewer for her/his valuable insights and constructive comments that have contributed to a better quality of our manuscript. Below are our responses in italics:
Responses to reviewer 2
- I thought this was a very valuable paper summarizing the perinatal management of pregnancies complicated by vWD in Slovenia. The content is quite interesting, but I think that the statistical analysis should use medians rather than means, as there must be a large skew in the distribution of the resulting data.
We are thankful for your aprreciation of our manuscript. Thank you for your observation and constructive comments. We agree with your opinion regarding statistic analysis of our data. Consequentially we repeated statistic analysis for nonparametric variables (gestational age, Apgar score) and used medians and interquartile range (IQR) for presentations of our results.
- Most of the Apgar scores, for example, would be 9 or 10. So, I would think that there would be no significant difference? Also, in Table 1, the presence of postpartum hemorrhage, blood transfusion, and bleeding due to perineal laceration are all correlated, which would explain the same thing, wouldn't it? We believe that the degree of anemia and platelet counts prior to delivery affect the postpartum values. Please reconsider this information as it affects the results of the paper.
As it was mentioned in the answer to previous question we have corrected statistic analysis for Apgar score and used Mann-Whitney U test for nonparametric variables. The p-value has not changed after the analysis (p<0,009) and the results are still significantly different. Nontheless, as the interquartile range is between 9-10 for women without VWD and 9-9 for women with VWD, most of the Apgar score values are 9 and 10. This finding is presumably the consequence of high number of controls (women without VWD), which causes that even clinically insignificant difference becomes statistically significant.
We agree with you that postpartum hemorrhage, received blood transfusions and bleeding due to perineal lacerations are correlated to some degree. On the other hand, in otherwise healthy women most cases of postpartum hemorrhage are due to uterine atony. Meanwhile in our study, there was no statistically significant difference regarding uterine atony between both groups (women with VWD and women without VWD). On the contrary women with VWD were more likely to experience childbirth trauma-related bleeding than women without VWD, which could be related to higher likelihood of postpartum hemorrhage in women with VWD. Additionaly not all women with primary postpartum hemorrhage (loss of > 500 ml of blood within 24 h after birth) require blood transfusion, as this decision depends on several factors, such as: complete blood count before and after delivery, amount of the blood loss during labor, possible symptoms and signs of anemia in women, etc.
- Please re-consider what might be prognostic factors in a patient with vWD.
Thank you for your question. If we are looking at a pregnant women with VWD risk for bleeding complications mostly depend on the type of VWD, with type I being mostly mild form of VWD and consequentially these patients rarely need prophylactic antihemorrhagic treatment. On the opposite, women with type 2 and type 3 are at higher risk for bleeding complications and almost allways need treatment such as plasma derived VWF and FVIII concentrates. Additionally, prognosis for pregnant women with VWD depends on the levels of VWF and FVIII and if this factors rise accordingly during pregnancy and before labor. Last but not least, it is of a great importance how levels of this factors fall during postpartum period, as their rapid fall represents high risk for postpartum hemorrhage. It is of great importance to measure levels of this factors before labor and for at least 5 days after the childbirth.
This manuscript is a resubmission of an earlier submission. The following is a list of the peer review reports and author responses from that submission.